# Management of Chronic Congestive Heart Failure Caused by Myxomatous Mitral Valve Disease in Dogs: A Narrative Review from 1970 to 2020

**DOI:** 10.3390/ani12020209

**Published:** 2022-01-16

**Authors:** Mara Bagardi, Viola Zamboni, Chiara Locatelli, Alberto Galizzi, Sara Ghilardi, Paola G. Brambilla

**Affiliations:** Department of Veterinary Medicine, University of Milan, Via dell’Università n. 6, 26900 Lodi, Italy; viola.zamboni@studenti.unimi.it (V.Z.); chiara.locatelli@unimi.it (C.L.); alberto.galizzi@unimi.it (A.G.); sara.ghilardi@unimi.it (S.G.); paola.brambilla@unimi.it (P.G.B.)

**Keywords:** chronic congestive heart failure, therapy, dogs, myxomatous mitral valve disease, narrative review

## Abstract

**Simple Summary:**

Myxomatous mitral valve disease (MMVD) is the most common acquired cardiovascular disease in dogs. The progression of the disease and the increasing severity of valvular regurgitation cause a volume overload of the left heart, leading to left atrial and ventricular remodeling and congestive heart failure (CHF). The treatment of chronic CHF secondary to MMVD in dogs has not always been the same over time. In the last fifty years, the drugs utilized have considerably changed, as well as the therapeutic protocols. Some drugs have also changed their intended use. An analysis of the literature concerning the therapy of chronic heart failure in dogs affected by this widespread degenerative disease is not available; a synthesis of the published literature on this topic and a description of its current state of art are needed. To the authors’ knowledge, a review of this topic has never been published in veterinary medicine; therefore, the aim of this study is to overview the treatments of chronic CHF secondary to MMVD in dogs from 1970 to 2020 using the general framework of narrative reviews.

**Abstract:**

The treatment of chronic congestive heart failure (CHF), secondary to myxomatous mitral valve disease (MMVD) in dogs, has considerably changed in the last fifty years. An analysis of the literature concerning the therapy of chronic CHF in dogs affected by MMVD is not available, and it is needed. Narrative reviews (NRs) are aimed at identifying and summarizing what has been previously published, avoiding duplications, and seeking new study areas that have not yet been addressed. The most accessible open-access databases, PubMed, Embase, and Google Scholar, were chosen, and the searching time frame was set in five decades, from 1970 to 2020. The 384 selected studies were classified into categories depending on the aim of the study, the population target, the pathogenesis of MMVD (natural/induced), and the resulting CHF. Over the years, the types of studies have increased considerably in veterinary medicine. In particular, there have been 43 (24.29%) clinical trials, 41 (23.16%) randomized controlled trials, 10 (5.65%) cross-over trials, 40 (22.60%) reviews, 5 (2.82%) comparative studies, 17 (9.60%) case-control studies, 2 (1.13%) cohort studies, 2 (1.13%) experimental studies, 2 (1.13%) questionnaires, 6 (3.40%) case-reports, 7 (3.95%) retrospective studies, and 2 (1.13%) guidelines. The experimental studies on dogs with an induced form of the disease were less numerous (49–27.68%) than the studies on dogs affected by spontaneous MMVD (128–72.32%). The therapy of chronic CHF in dogs has considerably changed in the last fifty years: in the last century, some of the currently prescribed drugs did not exist yet, while others had different indications.

## 1. Introduction

Myxomatous mitral valve disease (MMVD) is the most common acquired cardiovascular disease in dogs [1]. It is estimated that approximately 10% of dogs presented to primary care veterinary practices have heart disease and that MMVD is the most common among them (75% of canine cases of heart disease) [2,3]. Mitral regurgitation (MR) is the earliest hemodynamic event. The progression of the disease and the increasing severity of valvular regurgitation generate a volume overload of the left heart, leading to left atrial and ventricular remodeling and congestive heart failure (CHF) [1,4]. Particularly, remodeling secondary to MMVD is associated with characteristic histopathologic features, such as the expansion of the extracellular matrix with glycosaminoglycans and proteoglycans, the alteration of the valvular interstitial cell, and the attenuation or loss of the collagen-laden fibrosa layer. These lead to malformation of the mitral apparatus, biomechanical dysfunction, and mitral incompetence. Mitral regurgitation is the most common manifestation of MMVD and, in advanced stages, associated volume overload promotes progressive valvular regurgitation, left atrial and left ventricular remodeling, atrial tears, chordal rupture, and CHF, as stated before [5]. Dogs with MMVD present left-sided CHF clinical signs and a history of tachypnea, restlessness, respiratory distress, or cough [3]. Dogs with acute CHF need hospitalization, while patients with chronic CHF can be managed at home. The treatment of chronic CHF secondary to MMVD in dogs has not always been the same. In the last fifty years, the drugs utilized have considerably changed, and some have changed their intended use. The therapeutic protocols have also substantially changed over time. In 1970, the research and the literature were human medicine oriented, CHF in dogs was usually experimentally induced, and dogs were considered experimental animals. Later, the health and wellness of dogs have become primary endpoints; therefore, the research has been carried out on dogs as patients, and the therapy of CHF secondary to MMVD has been studied mainly on the spontaneous disease.

An analysis of the literature concerning the therapy of CHF in dogs affected by MMVD from 1970 to 2020 is not available, and a synthesis of the published literature on this topic, as well as a description of its current state of art, are needed.

The review on a specific subject can be systematic (SR) or non-systematic (narrative review—NR) and may include studies with different levels of evidence, different objectives, methods, and application areas [5]. Generally, an NR describes and appraises published articles that may be organized in chronological order, and the general framework includes different sections: introduction, literature search, central body/discussion, conclusion, and abstract.

To the authors’ knowledge, an NR on the therapeutic management of CHF secondary to MMVD in dogs has not yet been published in veterinary medicine, and the aim of this study is to overview the treatments of this disease in dogs from 1970 to 2020 using the general framework of narrative reviews.

## 2. Materials and Methods

### 2.1. Search Strategy

The identification of the research engines was the first step of the literature selection process [5]. The most accessible open-access databases, PubMed, Embase, and Google Scholar, were chosen, and the searching time frame was set in five decades, from 1970 to 2020. Then the key concept “therapy of chronic congestive heart failure caused by mitral valve disease in dogs” was transformed into keywords through the thesaurus system used to index articles for PubMed (Medical Subject Heading—MSH). The most distinctive terms were “therapy”, “chronic congestive heart failure”, “dogs”, and “mitral valve disease”, and they were entered into the search bar [5]. Then the categories of the most used cardioactive drugs (i.e., diuretics, ACE-inhibitors, inodilator, inotropes) were also included in the searching string.

The three basic Boolean operators, AND, OR, and NOT, were employed to focus the search. Since our topic contains multiple search terms, the operators were essential to connect various pieces of information to find exactly what we were looking for.

In accordance with the study design proposed by Ranganathan [6] and Röhrig, [7] the publications were classified as descriptive (case reports, case series, cross-sectional studies) or analytical, such as observational (cross-sectional studies, case-control studies, cohort studies) and experimental studies (clinical trials, randomized controlled trials, comparative studies, retrospective studies), or as questionnaires, reviews, and guidelines.

### 2.2. Inclusion/Exclusion Criteria

Inclusion criteria were full text and/or abstract available in the English language, better both, or at least the abstract, and studies focusing on the treatment of CHF secondary to MMVD in dogs published from 1970 to 2020. Furthermore, the selection also considered the origin of the CHF secondary to spontaneous or experimental-induced MMVD (MMVD^EI^).

Studies concerning the therapy of acute CHF, right CHF, Bernheim Syndrome and ventricular interdependence, chronic CHF secondary to heart diseases different from MMVD, research on animals different from the dog, and studies dealing with surgical therapy of MMVD were excluded from this study.

The inclusion and exclusion criteria were applied, and the resulting flowchart is reported in Figure 1.

The selected studies were subsequently classified into categories depending on the aim of the study, the population target, the pathogenesis of MMVD (natural occurring or induced), and the resulting CHF. Three sets of studies were obtained: dogs as a model for humans (DMH), dogs as a model for dogs (MMVD^EI^) (DMD), and dogs affected by naturally acquired MMVD as a model for diseased dogs (DNAD) (Table 1).

The studies were also distinguished depending on the target of the journal (human medicine and veterinary medicine) (Table 1) [8].

## 3. Results

### 3.1. General Considerations

The research engines provided 384 studies concerning the therapy of chronic CHF in dogs. PubMed proved to be more useful than Embase and Google Scholar for the search of older studies because it included more studies that better fitted our inclusion criteria (full text and/or abstract in English) (Figure 2 and Figure 3).

The studies published in veterinary medicine journals resulted in 43 (24.29%) clinical trials, 41 (23.16%) randomized controlled trials, 10 (5.65%) cross-over trials, 40 (22.60%) reviews, 5 (2.82%) comparative studies, 17 (9.60%) case-control studies, 2 (1.13%) cohort studies, 2 (1.13%) experimental studies, 2 (1.13%) questionnaires, 6 (3.40%) case-reports, 7 (3.95%) retrospective studies, and 2 (1.13%) guidelines (Figure 4).

In the fifty years considered, however, the experimental studies published in veterinary medicine journals (DMD) were less numerous (49–27.68%) than the studies on dogs affected by spontaneous MMVD (128–72.32%) (Figure 5).

Experimental studies in which the CHF was induced in healthy dogs to test the therapeutic efficacy (human or canine) show the highest timeframe variability in the distribution (Figure 6).

### 3.2. 1970–1979

In the 1970s, the literature search performed on the 3 research engines resulted in 11 studies, all of them published in human medicine journals [9,10,11,12,13,14,15,16,17,18,19]. The mitral valve incompetence was always experimentally induced in dogs, surgically and/or pharmacologically, to obtain a low cardiac output. The goal of these studies was to test drugs for the treatment of CHF in humans. The database that proved most useful for finding the oldest studies was PubMed (Figure 3).

In this decade, only one case report describing spontaneous CHF in dogs was published [20]. Thus, it can be inferred that pharmacologic management of canine CHF secondary to MMVD was not yet thoroughly investigated in this decade.

Furthermore, the milestone of therapy of the canine spontaneous chronic CHF due to MMVD was reported in a book [21]; therefore, it was not accessible to most of the researchers [18]. The therapeutic protocol included the same drugs used today but with very different intents, as reported in Figure 7 [21].

### 3.3. 1980–1989

During the 1980s, canine experimental studies increased compared to the 1970s [22,23,24,25,26,27,28,29,30,31,32,33,34,35,36,37,38,39,40,41,42,43,44,45,46,47,48,49,50,51,52,53,54,55,56,57,58,59], and they were mostly published in human medicine journals (DMH); in this decade, this type of study was still the most common (Figure 3). However, an experimental study carried out on dogs, that was then published in a veterinary medicine journal (DMD), was identified [60].

In addition to the above-mentioned experimental studies, five others were published on CHF management in dogs affected by spontaneous MMVD. Two studies tested the use of hydralazine [61,62], one tested milrinone [63] and the two others tested digitalis glycosides [64,65]. In the 1970s and 1980s, the digitalis glycosides were considered the standard therapy of the canine CHF, and the studies were mostly focused on finding the appropriate and individual therapeutic dosage; in fact, in some cases, the same dosage provided different results. For example, in one of the two publications, the digoxin dosage administered in 10 patients was 0.01 mg/kg lean body mass twice daily; however, serum digoxin concentrations in patients differed.

### 3.4. 1990–1999

The 1990s registered the highest number of experimental studies published in human medicine journals [66,67,68,69,70,71,72,73,74,75,76,77,78,79,80,81,82,83,84,85,86,87,88,89,90,91,92,93,94,95,96,97,98,99,100,101,102,103,104,105,106,107,108,109,110,111,112,113,114,115,116,117,118,119,120,121,122,123,124,125,126,127,128,129,130,131,132,133,134,135,136,137,138]. This highlights that the research concerning the therapy of human CHF was a key point along this period and that the dog was considered one of the most common and useful experimental animals. Furthermore, the number of experimental studies published in veterinary medicine journals also increased compared to the previous decade [139,140,141,142,143,144]. The efficacy of the therapy of CHF in dogs affected by spontaneous MMVD was evaluated, and the published studies were more numerous than in the previous twenty years. Many of the studies tested the effect and the efficacy of ACE inhibitors (captopril, enalapril, quinapril, ramipril, and benazepril), and over these ten years, they were introduced in the standard therapy of chronic CHF in dogs affected by MMVD [145,146,147,148,149,150,151,152,153,154,155,156]. In particular, it is important to mention the COVE study [147], the IMPROVE study [148], the BENCH study [156], and the LIVE study [155], which significantly contributed to the evaluation of the efficacy and effect of enalapril and benazepril. The ACE inhibitors were administered at different dosages: benazepril from a minimum of 0.25 mg/kg once daily, captopril 0.5 mg/kg three times daily, quinapril 0.5 mg/kg once daily, enalapril from 0.38 mg/kg twice daily to 0.5 mg/kg once or twice daily, and ramipril 0.125 mg/kg once or twice daily. Moreover, two studies have been published on milrinone [157,158] (which was administered at a dosage of 0.5 to 1 mg/kg twice daily), two on propentophylline [159,160], and one on the effect of fish oil when added to the diet of dogs with CHF [161]. These studies were isolated, and the research on these molecules has not been followed up in veterinary medicine. In 1991 and 1998, the first two systematic reviews, one on the management of chronic CHF in dogs [162] and one on afterload reducing agents [163], were published.

In 1995, the results of the first documented experience of a cardiological survey in veterinary medicine were published [164]. The questionnaire was submitted to veterinarians, and the aim was to analyze the preferences on the medications prescribed to treat heart diseases in dogs [164]. Interestingly, diuretics appeared to be the most common category (74%), while positive inotropic agents (digitalis glycoside), which were the milestone of the therapy of CHF in dogs up to the 1980s, were the least used drugs (20%). Thus, from this decade on, the use of digitalis glycoside as the main therapeutic agent started decreasing [164].

### 3.5. 2000–2009

During the 2000s, the experimental studies that tested in dogs the efficacy of drugs used for CHF in humans (DMH) were fewer than in the previous decades [165,166,167,168,169,170,171,172,173,174,175,176,177,178,179,180,181,182,183,184,185,186,187,188,189,190,191,192,193,194,195,196,197,198,199,200,201,202,203,204,205,206,207,208,209,210,211,212,213,214,215]. In contrast, there was an increase in studies published in veterinary medicine journals that evaluated the treatment of CHF in dogs, both experimental [216,217,218,219,220,221,222,223,224,225,226,227,228,229,230,231] and carried out in patients with spontaneous MMVD. The scientific publications on drugs tested in dogs with CHF secondary to MMVD (DNAD) were more numerous than in previous years, and many studies were performed on ACE inhibitors [232,233,234,235], β-blockers [236,237,238], sildenafil [239,240], amiodarone [241], diuretics [242], isosorbide 5-mononitrate [243], and amlodipine [244]. In particular, the different ACE inhibitors were administered at different dosages: quinapril and enalapril at 0.5 mg/kg once daily, ramipril at 0.125 mg/kg once daily, perindopril at 0.2 mg/kg once daily, benazepril from 0.25 to 1 mg/kg/day. Pimobendan was also introduced in the standard therapeutic protocol of CHF secondary to MMVD [245,246,247,248,249]. Some studies compared pimobendan and ACE inhibitors, such as the QUEST study [250], and the results showed that pimobendan, in combination with the standard therapy (diuretics and digoxin), administered at a dosage of 0.4–0.6 mg/kg/day prolonged the expectancy and quality of life in patients affected by CHF secondary to MMVD [251]. Digitalis glycosides definitively changed their intended use, and, differently from their previous indication, they were administered to decrease the excitability of the atrioventricular node and the ventricular rate in presence of atrial fibrillation (AF) [252]. At the end of this decade, guidelines for the diagnosis and treatment of canine chronic valvular heart disease were published for the first time [2]; the recommended drugs were furosemide, ACE-I, and pimobendan. The majority of the panelists recommended the administration of β-blockers, digoxin, or diltiazem as antiarrhythmics, and cough suppressants and bronchodilators in the presence of cough. In this decade, the attention to dogs’ diets increased, with particular interest for sodium restriction [253,254,255,256,257]. The reviews published in these ten years are much more numerous than in previous years because more studies were carried out to evaluate the therapy of CHF secondary to MMVD in dogs [237,247,258,259,260,261,262,263,264]; these data are important because the reviews identify, evaluate, and summarize the outcomes of the studies carried out in a specific period.

### 3.6. 2010–2020

From 2010 to 2020, as well as in the early 2000s, there was a decrease in experimental studies carried out on dogs published in human medicine journals and an increase in studies published in veterinary medicine journals. DMH studies were 34 [265,266,267,268,269,270,271,272,273,274,275,276,277,278,279,280,281,282,283,284,285,286,287,288,289,290,291,292,293,294,295,296,297,298], the DMD studies were 25 [299,300,301,302,303,304,305,306,307,308,309,310,311,312,313,314,315,316,317,318,319,320,321,322,323,324], and 69 studies concerning the therapy of CHF secondary to spontaneous MMVD have been found in the literature. The research was profuse, and the drugs tested were numerous; in fact, the reviews published in these ten years regarding the therapy of CHF secondary to MMVD in dogs are 25 [325,326,327,328,329,330,331,332,333,334,335,336,337,338,339,340,341,342,343,344,345,346,347,348,349]. As in the previous decade, many studies have been published and many drugs have been tested on dogs with spontaneous disease, such as sildenafil [350,351,352,353], imatinib [354], atorvastatin [355,356], coenzyme q10 [357,358], amlodipine [359], BNP1–32 [360], and angiotensin receptor antagonists [361]. In this decade, several studies have been published on diuretics, with particular regard to torasemide (such as the TEST study [362] and the CARPODIEM study [363]). In these papers, the diuretic has been administered over a dose range of 0.13 mg/kg/day to 0.5 mg/kg/day, and spironolactone (an aldosterone receptor antagonist) has been administered at dosages from 0.49 mg/kg once daily to 2 mg/kg once daily. These drugs are very useful in the therapy of CHF secondary to MMVD in dogs [362,363,364,365,366,367,368,369]. 

Studies focused on other standard therapy drugs (pimobendan or ACE inhibitors, such as the QUEST and the EPIC studies [370,371], respectively, which focused on symptomatic and asymptomatic dogs with cardiac remodeling) or drugs used in case of complications related to this syndrome (such as β-blockers, amlodipine, and digoxin) [372,373,374,375,376,377,378,379,380,381,382,383,384] have also been published. In the QUEST and EPIC studies, pimobendan was administered over a dose range of 0.4 to 0.6 mg/kg/daily and benazepril over a dose range of 0.25 to 1 mg/kg/daily. The addition of the ACE inhibitor ramipril to pimobendan and furosemide has been demonstrated not to have any beneficial effect on survival time in dogs with CHF secondary to MMVD [384].

In 2019, the newly updated guidelines were published according to the most recent results [3]. The recommended diuretic was not only furosemide but also torasemide. Moreover, spironolactone was recommended in addition to the classic triple therapy for its aldosterone-antagonist effect [3].

Other studies evaluated the clinical findings and the survival time following the administration of the drugs [385], X-ray and ultrasound images after the treatment [386], and the “aldosterone breakthrough” following the administration of ACE inhibitors [387]. Furthermore, a study on a myostatin antagonist used to counteract cardiac cachexia secondary to CHF has also been published [388]. In recent years, the evaluation of NT-pro BNP levels following the cardiological treatment has gained a lot of interest; in fact, low levels of this molecule indicate a better prognosis [389,390]. Cohort studies in veterinary medicine are rarely carried out because of high costs and long-time needing; in the period considered for this narrative review, only two have been found in the literature [391,392]. A questionnaire published in 2015 was also very interesting since it is useful to compare how dogs’ CHF therapy has changed from that which was used twenty years earlier [393].

### 3.7. 2021—Last Minute Update

The never-ending story of the medical strategies for the treatment of MMVD in dogs is far from the conclusion.

In 2021, the last clinical trial, named the BESST study, has been published after the conclusion of this narrative review [394]. The BESST is a multicentric double-blind study that compared the combination of benazepril and spironolactone in the management of CHF in dogs affected by MMVD [394]. Particularly, the results of this study show that the combination of spironolactone and benazepril is effective, safe, and superior to benazepril alone when used with furosemide for the management of mild, moderate, or severe CHF caused by MMVD in dogs.

## 4. Discussion

The therapy of chronic CHF in dogs has considerably changed in the last fifty years. In the last century, some of the currently prescribed drugs did not exist yet, while others had different indications [21]. Digitalis glycosides are the oldest medication used for the treatment of chronic CHF in dogs, and despite the risks of intoxication or death secondary to their overdose, the drug is still used. For a very long time, digitalis glycosides have been administrated because of their positive inotropic and negative chronotropic effect, and their dromotropic action. Furthermore, they were used to relieve clinical signs as cough, dyspnea (secondary to pulmonary edema), and ascites, frequently found in severe CHF. Currently, their clinical use has changed: the presence of AF, quite a common complication of MMVD, especially in medium- and large-breed dogs, is the first indication of adding the digitalis glycosides to the therapeutic protocol. Nevertheless, digoxin is the most common drug used to ensure adequate control of the ventricular rate in patients with AF, alone or associated with diltiazem, as reported by the literature [252,377].

Diuretics are another pivotal category of drugs that makes the history of the cardioactive protocols. In 1970, diuretics were administered only if therapy with glycosides alone failed. Nowadays, they are one of the most useful drugs used for the treatment of chronic CHF in dogs. The oldest diuretics were mercurials, thiazides, furosemide, spironolactone, and ethacrynic acid. Furosemide was administered in case of the development of resistance to thiazides, so it was not the first diuretic of choice [348]. To the present day, the therapeutic protocol of chronic CHF in dogs includes furosemide, the most widely used diuretic.

Interestingly, the role of spironolactone in cardiac protocols changed its use: in the last century, it was administered in association with thiazides to avoid hypokalemia, while it is now one of the main drugs for the treatment of chronic CHF because of its antagonist activity to mineralocorticoid receptors. Spironolactone, added to the therapy with pimobendan, diuretic, and an ACE inhibitor, has been shown to increase survival time in dogs with chronic CHF secondary to MMVD [362]. The main antiarrhythmics drugs used in the 1970s were quinidine sulphate, procainamide, and lidocaine. Currently, as previously mentioned, the digoxin–diltiazem protocol is well tolerated, and β-blocker antiarrhythmics can be used [215,219]. ACE inhibitors were introduced into dogs’ treatment protocol in the early 1990s, while pimobendan was introduced in the 2000s.

The renin–angiotensin–aldosterone system (RAAS) activation can be compensatory in the early stages of cardiovascular and renal diseases, but its long-term activation is maladaptive [345]. In patients with heart failure, relative increases in plasma renin activity and the blood aldosterone concentration are considered markers of, and contributors to, the hemodynamic and anatomic derangements of this syndrome [345]. However, the literature shows that the more we learn about this system, the broader and more complex it becomes. Continuous research into this complex system is necessary to improve medical therapies for cardiovascular and renal diseases, allowing us to modulate this system and improve clinical outcomes more adeptly [345].

Following this research, other considerations emerge: fifty years ago, the research was mainly focused on the therapy of human CHF and, in the studies published in human medicine journals that tested drugs on dogs, the syndrome was induced in different ways, surgically or pharmacologically. Different pathologies that cause CHF, such as the surgical procedure or the drug used to induce a low cardiac output, were not particularly taken into consideration.

Studies published in veterinary journals were only a few in the 1970s; however, over the years, their number increased until they were more than the studies published in human journals where the dog was used as a laboratory animal. Furthermore, regarding the studies published in veterinary medicine journals, it should be noted that the experimental studies (DMD) are fewer than the studies carried out on dogs with spontaneous pathology.

As can be seen from Figure 2, over the years, the types of studies have increased considerably in veterinary medicine, and reviews have always had an increasing trend over the five decades. It is important to remember that reviews should be published every 4–5 years as they are useful to understand the direction of the research; however, systematic reviews are restricted in veterinary medicine. Cohort studies are not very numerous in veterinary medicine due to their high cost, long duration, and potential numerous losses of subjects during the study. Over the past two decades, randomized controlled trials have become more and more numerous; they have the advantage of selecting a group of patients through defined criteria, administering the treatment randomly, and thus reducing bias.

The databases chosen for the research of the studies to be included in this narrative review were PubMed, Google Scholar, and Embase. PubMed was the database that provided the greatest number of studies and, in particular, the oldest ones. It is a very wide database and, thanks to the use of Mesh terms and Boolean operators, allows the researcher to identify the most pertinent studies easily and quickly. Google Scholar is a very intuitive and easy-to-use database. The results obtained are conspicuous as there are also books, citations, meetings, and symposiums; however, it is more generic compared to PubMed, and sometimes some results are irrelevant. It is important to remember that Google Scholar is the only free access database among these three; not all information may be accessible without an institutional account, but it allows anyone to search. Embase is a database that allows precise searches, although fewer old studies have been found compared to PubMed.

This review is a narrative review (NR) or non-systematic review: NRs are aimed at identifying and summarizing what has been previously published, avoiding duplications, and seeking new study areas that have not yet been addressed. An NR does not have the strict rules of a systematic review; therefore, an NR potentially leads to biases because of subjectivity in the study selection [5].

## 5. Conclusions

The treatments of chronic CHF secondary to MMVD in dogs have considerably changed in the last fifty years, and some drugs have changed their intended use, such as digitalis glycosides and spironolactone. In the 1970s, the research and the literature were human medicine oriented, CHF in dogs was usually experimentally induced, and dogs were considered experimental animals. Later, the health and wellness of dogs have become primary endpoints; therefore, the research has been carried out on dogs as patients, and the therapy of CHF secondary to MMVD has been studied mainly on the spontaneous disease. This NR has been aimed at identifying and summarizing what has been previously published, avoiding duplications, with the intent to be a useful tool for the clinicians approaching this topic.

## Figures and Tables

**Figure 1 animals-12-00209-f001:**
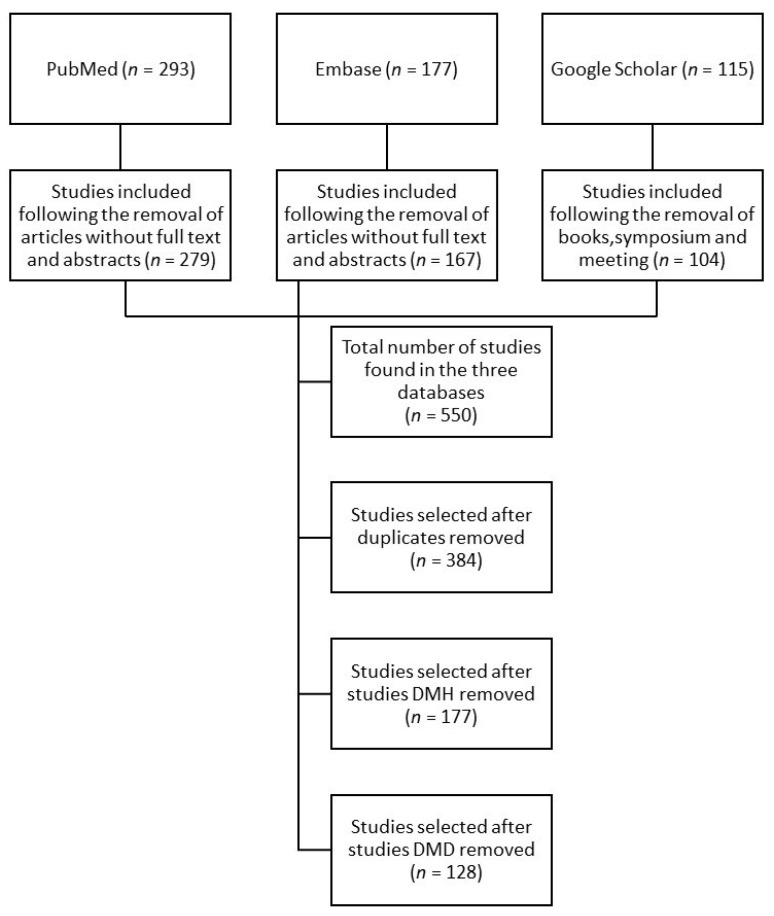
Flow chart of the literature selection process for the present article.

**Figure 2 animals-12-00209-f002:**
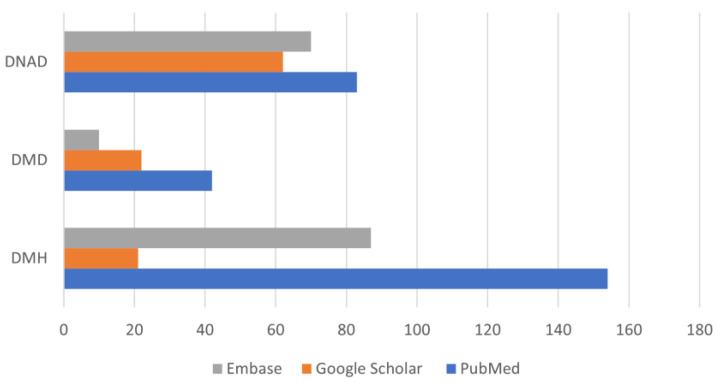
Number of studies identified in the three databases and allocated into three categories.

**Figure 3 animals-12-00209-f003:**
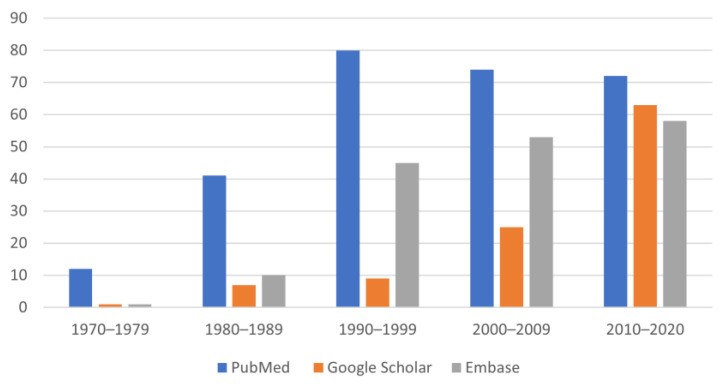
Numerousness of studies on CHF in dogs found in each database and in all decades.

**Figure 4 animals-12-00209-f004:**
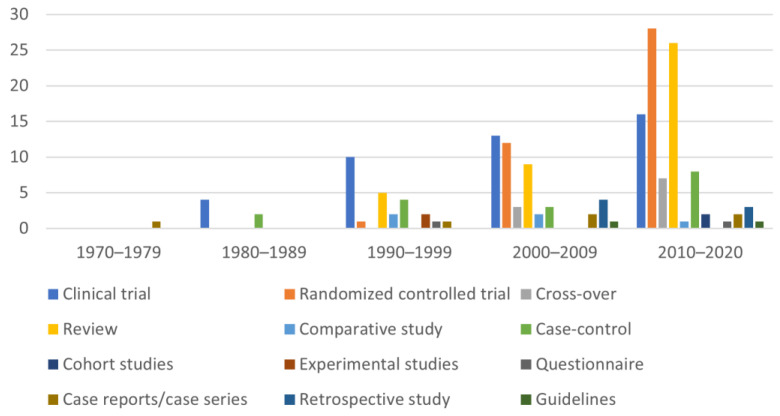
Subdivision of the studies published in veterinary medicine journals (DMD and DNAD) according to the type of study in the five decades taken into consideration. It is interesting to note that in the last decades the typologies of veterinary medicine studies were more diversified than in the last century.

**Figure 5 animals-12-00209-f005:**
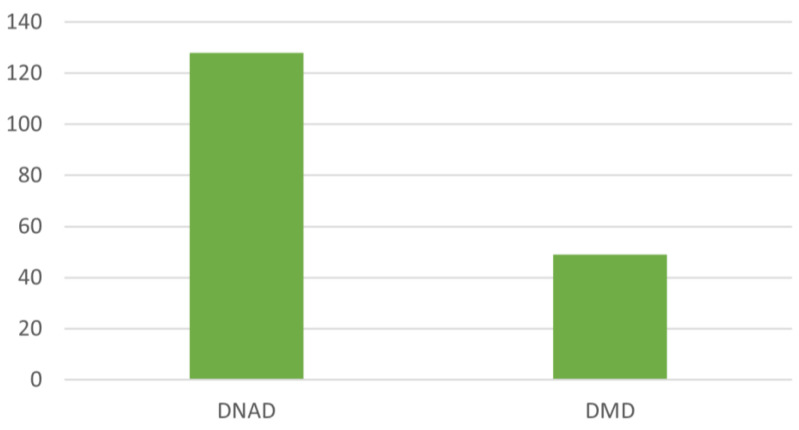
Comparison of numerous studies published in veterinary medicine journals.

**Figure 6 animals-12-00209-f006:**
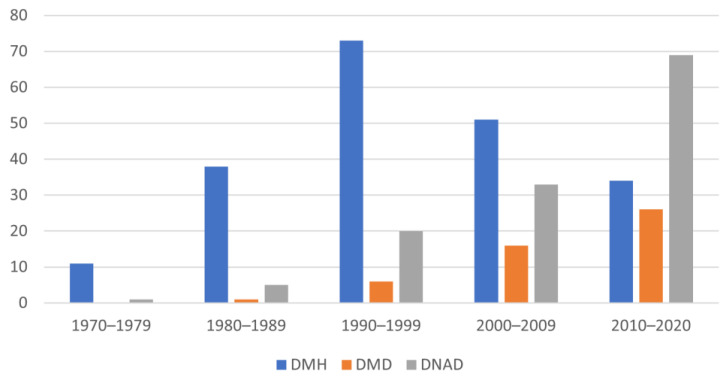
Studies found in each database, allocated into three categories, and distributed in the considered decades. It is interesting to note the increased number of studies carried out on dogs with a spontaneous disease in the last decades compared to studies in which the dog was a model for testing drugs potentially useful for the treatment of human CHF.

**Figure 7 animals-12-00209-f007:**
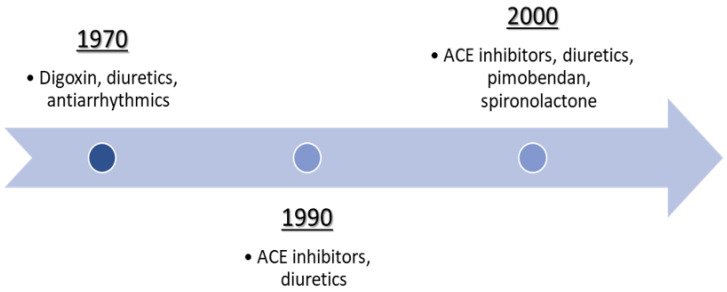
Timeline representing the main categories of drugs used for canine CHF therapy over the past fifty years.

**Table 1 animals-12-00209-t001:** Subclassification of the studies into three categories.

	DMH	DMD	DNAD
Journal	Human medicine journal	Veterinary medicine journal	Veterinary medicine journal
Aim of the study	To test efficacy of medication of CHF in humans	To test efficacy of medication of CHF in dogs	To test the therapy of CHF in dogs
Population	Healthy dogs	Affected or healthy dogs	Affected dogs
MMVD	Induced	Natural or induced	Natural

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
