# Peer review of "Management of Chronic Congestive Heart Failure Caused by Myxomatous Mitral Valve Disease in Dogs: A Narrative Review from 1970 to 2020"

_animals, 2022, doi:10.3390/ani12020209_

Round 1

Reviewer 1 Report

Please add the doses of drugs used in clinical trialsin paragraphs 3.3-3.6

Please consider adding a paragraph from a clinical trial from 2021  (VALVE and BESST trials) 

Author Response

Reviewer 1

Please add the doses of drugs used in clinical trialsin paragraphs 3.3-3.6

Please consider adding a paragraph from a clinical trial from 2021 (VALVE and BESST trials) 

Thank you very much for your suggestions. We specified the dosages of the drugs in the paragraphs indicated by Reviewer 1 and include a sentence that describes the VALVE study results. Furthermore, we added a paragraph named “Last minute publications” to better describe the BESST trial. We decided not to include this reference in the previous paragraphs because this paper has been published in 2021 and our manuscript is a review of literature until 2020.

----------------------------------------------------------------------------------------

The authors thank the Reviewers for the thorough review of our study. We feel that the quality of our manuscript has improved following their comments and suggestions.

Thank you very much.

Best regards

Mara Bagardi

Reviewer 2 Report

The topic is interesting. The development turns out to be generic, perhaps too generic. It is appropriate to deepen the work and enhance the selected bibliographic references.

Below are the points on which a comment on the proposed text is needed.

  • Line 13 - Ventricular remodeling, but in general of the heart chambers, is an important phenomenon in the CHF subject of this review but it is treated quickly and superficially. The authors report twenty articles on remodeling, but the arguments are sparse and superficial. Please, it is appropriate to reconsider this part.
  • Line 29. The authors write that they have selected 384 articles, but the articles reported in the appropriate section of the article are 394. Is it possible to verify this step?
  • Line 83. The main books on veterinary pharmacology, pharmacotherapy and cardiology have not been used in this review. Can the authors comment on the motivations?
  • Line 102. Can the authors explain why an abstract may be sufficient to obtain information? have the authors adopted an additional criterion to select abstracts?
  • Line 106. No articles on secondary involvement of the right heart in case of MMVD of mitral valve have been examined, despite its involvement may play an important role in chronic forms. Can the authors explain, comment, and reconsider this part?
  • Line 135. Articles on experimental studies have been selected. Considering the concept of chronicity, how overlapping are they? how useful can they be to understand the natural history and therapy of MMVD?
  • Line 165. Why shouldn't a book be accessible to researchers? and what would this book be?
  • Why use 384 / 394 publications without mentioning them all in the text?

Author Response

Reviewer 2
The topic is interesting. The development turns out to be generic, perhaps too generic. It is 
appropriate to deepen the work and enhance the selected bibliographic references.
Below are the points on which a comment on the proposed text is needed.
• Line 13 - Ventricular remodeling, but in general of the heart chambers, is an important 
phenomenon in the CHF subject of this review but it is treated quickly and superficially. The 
authors report twenty articles on remodeling, but the arguments are sparse and superficial. 
Please, it is appropriate to reconsider this part.
Thank you for you comment. We have added information about remodeling at lines 51-59. We 
have not modified the summary due to the low number of words allowed for this section. 
• Line 29. The authors write that they have selected 384 articles, but the articles reported in 
the appropriate section of the article are 394. Is it possible to verify this step?
Thank you for this comment. Does the Reviewer refer to the studies reported in the reference 
section? In this case we underline that in that section we also reported the studies used as a 
reference for the drafting of the introduction section (relatively to the natural history of MMVD 
and to the narrative review creation). This is the reason for the higher number in the reference 
section.
• Line 83. The main books on veterinary pharmacology, pharmacotherapy and cardiology 
have not been used in this review. Can the authors comment on the motivations?
Thank you. We have considered some cardiology books in our reference list, as reported in the 
reference section. The authors decided to use the most common-used databases and did not 
consider the books on veterinary pharmacology and pharmacodynamics because it was not among 
their aims. We also believe that the review would have been far too far-reaching, if we would have 
further expanded the bibliographical sources including books regarding very specific issues.
• Line 102. Can the authors explain why an abstract may be sufficient to obtain information? 
have the authors adopted an additional criterion to select abstracts?
Thank you for this comment. Abstracts are sufficient to obtain information about the type of the 
study and allowed us to correctly classify them in the described categories. We did not adopt an 
additional criterion to select abstracts. 
• Line 106. No articles on secondary involvement of the right heart in case of MMVD of 
mitral valve have been examined, despite its involvement may play an important role in 
chronic forms. Can the authors explain, comment, and reconsider this part?
Thank you for this comment. We confirm that we decided not to include studies describing right 
CHF and we have also specified that the description of Bernheim Syndrome and the ventricular 
interdependence were not in our focus. 
• Line 135. Articles on experimental studies have been selected. Considering the concept of 
chronicity, how overlapping are they? how useful can they be to understand the natural 
history and therapy of MMVD?
Thank you for this comment. Experimental studies, as reported at line 98, are clinical trials, 
randomized controlled trials, comparative studies, retrospective studies. We think that these types 
of studies consider the concept of chronicity and help to understand the natural history and therapy 
of MMVD. If the Reviewer refers to experimentally-induced MMVD instead, a great number of 
these studies compare the effect of the selected drugs both in acute and chronic CHF or the longterm effect of the drug in the dog. Other studies report the effect of the drug in the dog considered 
as a model for human and as a model for chronic treatment of CHF (a lot of studies were carried 
out on inotropes, beta-blockers or ACE inhibitors). 
• Line 165. Why shouldn't a book be accessible to researchers? and what would this book be?
Thank you very much for this comment. This was a mistake. We have corrected it. 
• Why use 384 / 394 publications without mentioning them all in the text?
Thank you very much. We are not sure if this is what the Reviewer means with this question, but we 
carefully checked the paper after this comment, and all the references reported in the reference 
section have a citation in the text. 
----------------------------------------------------------------------------------------
The authors thank the Reviewers for the thorough review of our study. We feel that the quality of 
our manuscript has improved following their comments and suggestions.
Thank you very much.
Best regards
Mara Bagardi

Reviewer 3 Report

The authors reviewed the literature from the past 50 years on the management of chronic heart failure in dogs secondary to myxomatous mitral valve disease in a narrative manner. It is indeed an interesting and relevant area to study and such an extensive literature analysis has not yet been conducted. Thus, the manuscript is of general interest and worth publishing. 

The number of references used is adequate to the topic. It is also worth noting that the authors specified search terms and explained types of literature included. English language throughout the manuscript is of high quality. Only few clarifications and corrections are needed.  

I would like the authors to address these minor issues: 

Lines 36-37: In order to unify presented data, please give the exact number (along with percentage) of experimental studies - as it is presented in the previous sentence and in Lines 142-143. 

Lines 66-75: In my opinion, it is not necessary to write in such detail about what a narrative review is. This section should be shortened.

Line 346: In 1970 -> In the 1970s - the reference is to an entire decade, not a specific year.

In the Discussion section, in the first three paragraphs you have addressed four major groups of drugs: digitalis glycosides, diuretics, spironolactone and antiarrhythmics, while only briefly mentioning the role of ACE-inhibitors in CHF therapy. I would like to see another paragraph discussing their role throughout these years.

Author Response

Reviewer 3

The authors reviewed the literature from the past 50 years on the management of chronic heart failure in dogs secondary to myxomatous mitral valve disease in a narrative manner. It is indeed an interesting and relevant area to study and such an extensive literature analysis has not yet been conducted. Thus, the manuscript is of general interest and worth publishing. 

The number of references used is adequate to the topic. It is also worth noting that the authors specified search terms and explained types of literature included. English language throughout the manuscript is of high quality. Only few clarifications and corrections are needed.  

I would like the authors to address these minor issues: 

Lines 36-37: In order to unify presented data, please give the exact number (along with percentage) of experimental studies - as it is presented in the previous sentence and in Lines 142-143. 

Thank you very much for this suggestion. We have added the number.

Lines 66-75: In my opinion, it is not necessary to write in such detail about what a narrative review is. This section should be shortened.

Thank you for your suggestion. We have deleted some useless information from this paragraph.

Line 346: In 1970 -> In the 1970s - the reference is to an entire decade, not a specific year.

Thank you for this comment. We have corrected this mistake.

In the Discussion section, in the first three paragraphs you have addressed four major groups of drugs: digitalis glycosides, diuretics, spironolactone and antiarrhythmics, while only briefly mentioning the role of ACE-inhibitors in CHF therapy. I would like to see another paragraph discussing their role throughout these years.

Thank you for this suggestion. We have added a little paragraph regarding the role of ACE inhibitors in CHF therapy. Thank you.

----------------------------------------------------------------------------------------

The authors thank the Reviewers for the thorough review of our study. We feel that the quality of our manuscript has improved following their comments and suggestions.

Thank you very much.

Best regards

Mara Bagardi

Reviewer 4 Report

The only comment of the reviewer is the suggestion to remove the penultimate sentence from the article.

Author Response

Reviewer 4

The only comment of the reviewer is the suggestion to remove the penultimate sentence from the article.

Thank you very much for your comment. We have deleted the penultimate sentence from the conclusion section as you suggested (“It is important to remember that reviews of literature should be published every 4-5 years as they are useful to understand the direction of the research”).

----------------------------------------------------------------------------------------

The authors thank the Reviewers for the thorough review of our study. We feel that the quality of our manuscript has improved following their comments and suggestions.

Thank you very much.

Best regards

Mara Bagardi

Round 2

Reviewer 2 Report

The authors made significant changes and clarified some doubts.